# Decontamination Efficiency of Thermal, Photothermal, Microwave, and Steam Treatments for Biocontaminated Household Textiles

**DOI:** 10.3390/molecules27123667

**Published:** 2022-06-07

**Authors:** Branko Neral, Selestina Gorgieva, Manja Kurečič

**Affiliations:** Faculty of Mechanical Engineering, University of Maribor, Smetanova Ulica 17, SI-2000 Maribor, Slovenia; selestina.gorgieva@um.si (S.G.); manja.kurecic@um.si (M.K.)

**Keywords:** household textiles, thermal treatment, decontamination, reduction efficiency

## Abstract

With the outbreak of the COVID-19 pandemic, textile laundering hygiene has proved to be a fundamental measure in preventing the spread of infections. The first part of our study evaluated the decontamination efficiency of various treatments (thermal, photothermal, and microwave) for bio contaminated textiles. The effects on textile decontamination of adding saturated steam into the drum of a household textile laundering machine were investigated and evaluated in the second part of our study. The results show that the thermal treatment, conducted in a convection heating chamber, provided a slight reduction in efficiency and did not ensure the complete inactivation of *Staphylococcus aureus* on cotton swatches. The photothermal treatment showed higher reduction efficiency on contaminated textile samples, while the microwave treatment (at 460 W for a period of 60 s) of bio contaminated cotton swatches containing higher moisture content provided satisfactory bacterial reduction efficiency (more than 7 log steps). Additionally, the treatment of textiles in the household washing machine with the injection of saturated steam into the washing drum and a mild agitation rhythm provided at least a 7 log step reduction in *S. aureus*. The photothermal treatment of bio contaminated cotton textiles showed promising reduction efficiency, while the microwave treatment and the treatment with saturated steam proved to be the most effective.

## 1. Introduction

The process of caring for household textiles usually consists of the following phases: laundering, drying, ironing, and storage. The primary purpose of caring for household textiles is to ensure the highest level of textile cleanliness and hygiene without changes in the textiles’ basic functions, such as comfort, functionality, and health protection.

Very often, it is forgotten that the conditions of the care process (e.g., temperature, time, water ratio/degree of humidity, concentration of detergent/bleaching agent/disinfectant, and mechanical action) must adapt to the physical–chemical characteristics of the textiles, such as the type of fibre and form (knitwear/fabric/nonwoven), the type of finish (bleached/dyed/printed/surface-functionalised), the level of hygroscopicity/hydrophobicity, the thermal conductivity, and the fastness type and level, and not the opposite. It seems to be entirely expected that the textile care process will not cause damage or shorten the life cycle of textiles and be environmentally sustainable. According to Bloomfield, textile laundering and drying are the most frequent household occupations, essential to ensuring hygiene, reducing the risk of infections, and, thus, protecting the health and safety of household members [1].

Statistics show that 82.5% of EU-28 households have a laundering machine, which is used 3.25 times/week, and 78.8% of households have a tumble dryer, which is used 2.05 times/week [2,3,4]. The annual energy consumption and water consumption of household laundering and drying machines in the EU-28 were estimated to be 35.3 TWh and 2.496 million m^3^, respectively [5]. The European Environmental Policy, which rests upon the precautionary and prevention principles, projected annual electricity savings of 2.5 TWh, leading to GHG emission reductions of 0.8 Mt CO_2_ eq/year and estimated water savings of 711 million m^3^ in 2030 [6]. Clearly, EU institutions have public support for their ambitious plan [7].

In recent years, the number of studies investigating household laundering technology has been increasing [8,9,10,11]. Several studies have found that reducing the laundering time, the bath temperature, and water and electricity consumption may have positive financial impacts (a reduction in household textile laundering costs) and ecological impacts (a reduction in environmental contamination) at a comparable cleaning efficiency. However, most of these studies overlook the physical and chemical characteristics of different textiles, such as relations between factors of the Sinner circle, which determine the laundering quality.

In contrast, several studies have shown that laundering household textiles at temperatures below 40 °C, with a low bath ratio and a short laundering time, and without bleaching/disinfection agents may cause problems with the hygiene of textiles and provide low decontamination efficiency. Furthermore, the results indicate that the washing machine itself may serve as a vector for the contamination of textiles, the formation of biofilms, the development of unpleasant odours, and the growth of pathogenic microorganisms [12,13,14].

Overall, these studies demonstrate that the removal of bacteria, fungi, and viruses from textiles during the care process is influenced decisively by the mechanical and chemical mechanisms related to temperature, loading ratio, addition of bleaching agents, and duration [15,16,17]. It is also important to note that, while textile washing temperatures below 50 °C remove 95% of microorganisms [18,19], the rest are expected to be destroyed by tumble drying and ironing (particularly steam ironing).

However, far too little attention has been paid to textile drying and decontamination processes. Only a few research reports have been published that refer to industrial or household textile drying and decontamination.

According to Carr, the term ‘drying’ is generally used to describe the process of dewatering textiles, either mechanically (squeezing, vacuum extraction, centrifugation) or thermally (conduction, convection, infra-red, microwaves). Thermal textile drying is a complex process affected by the thermal and sorption properties of textiles, heat and mass transfer to and from the surface and inside of the textile fibre, the hydrodynamics of particle motion inside the fibre, and the various mechanisms by which moisture migrates through porous textile materials. The traditional thermal textile drying process is extremely energy-intensive and must be planned out and performed thoroughly [20,21,22].

Thermal textile drying in industrial laundries is performed with a drum dryer, a garment finisher, or a flatwork ironer depending on the form, the fibre type, and the surface functionalisation of the textile. Household textiles are usually dried in one of two ways: enhanced outdoor drying with wind and sunlight or indoor drying with a tumble dryer or drying cabinet [23,24,25].

For both industrial and household textile drying, the most critical parameter in the correctly performed process is the amount of moisture inside the textile material during, and especially at the end, of the drying process (moisture regain). Under-drying may result in the growth of microorganisms, whilst over-drying may result in a reduction in hygroscopic moisture that may cause a change in the fibre’s morphology and, thus, a deterioration in product quality [21,26]. These factors can, consequently, shorten the lifespan of household textiles and, at the same time, increase emissions of textile wastes.

Upon analysing a few studies, Bloomfield found that textile drying after laundering can reduce the microbial load by between 0.4 and 4.5 log colony-forming units (CFUs), despite the significant variations in the data. The results suggest that drying at higher temperatures can produce further reductions in the numbers of microorganisms that survive the laundering process. Bloomfield also found that another factor likely to affect microorganisms’ detachment is the extent to which contaminated fabrics are dried before laundering [27]. Together, the studies reviewed in the IFH 2011 Report show that the strength of microorganisms’ adhesion to fabrics increases with drying [28].

The textile industry has investigated many uses for microwave energy, from heating, drying, dye-fixation, and the curing of resin-finished fabrics to disinvesting wool fabrics, but only from the heating point of view [29,30,31,32,33]. It can be concluded that significant progress has been made in the development and application of microwave technology in textile finishing processes, while its applicability to decontamination has been more or less overlooked.

Chen studied the applicability of a kitchen microwave oven for the decontamination of cotton inoculated with *Aspergillus niger,* a common mould (a type of fungus) that lives indoors and outdoors and can cause people with a weakened immune system to have health problems (e.g., allergic reactions, lung infections, and infections in other organs). The results of studies indicate that microwave irradiation has potential as a tool for textile decontamination; however, the limitations of conventional household microwave ovens (e.g., low efficiency, nonuniform heating, and a lack of a continuous source of moisture) need to be assessed, and at least partially rectified, in order to render microwave treatment a viable and practical tool for textile decontamination [34,35,36].

Gupta and Versteeg reviewed thirty-three studies from 1920 to 2016 to provide insight into footwear, sock, and textile sanitisation. They evaluated less-known techniques, such as UV light, silver-light (silver oxide (Ag_2_O) activated by ultraviolet light), and ozone. The authors found that new approaches to shoe and sock sanitisation have yielded auspicious results, but further research and development are needed [37].

SARS-CoV-2 has spread to all continents in just a few weeks. It has been reaffirmed that basic hygiene measures, including textile hygiene, are the most effective non-pharmacological measures in limiting the spread of viruses [38,39,40,41,42]. Unfortunately, one of the features of the first wave of the COVID-19 pandemic was the lack of protective face masks and timely and accurate recommendations on the care and disinfection of textiles. Hospital and industrial laundries adapted their washing and disinfection processes quickly to the new conditions provided by the manufacturers of industrial detergents and disinfectants. However, significantly less information and fewer recommendations were available to households.

Bloomfield and Bockmühl concluded that textile laundering is the most frequent household occupation, essential to ensuring hygiene, reducing the risk of infection, and, thus, protecting the health and safety of household members [1,43]. This finding can also be linked to statistical data on SARS-CoV-2 infections in the Republic of Slovenia collected by the National Institute of Public Health from 1 November 2020 to 19 March 2022. Figure 1 shows the weekly number of infections depending on the most probable location of virus transmission, e.g., household, industry, hospital/healthcare institution, and nursing home.

Figure 1 shows that the number of likely infections in workplaces and households at the end of 2020 was almost equal, slightly lower in nursing homes, and the lowest in hospitals. When the lockdown was introduced, the number of likely infections in workplaces fell sharply, in contrast to households, where it began to rise and then finally subsided in the first quarter of 2022.

In this study, we focused on less-investigated procedures for removing microbial contaminants from household textiles. In the first part of the study, we investigated and evaluated the effects of microorganism decontamination on textiles by the application of hot air, a combination of thermal and UV drying (photothermal treatment), microwaves, and saturated steam at atmospheric pressure. Analysis of the obtained results helped us to specify the conditions under which we can achieve the highest level of decontamination of textiles before or after household textile laundering and tumble drying.

## 2. Results and Discussion

The main objective of this research was to evaluate the effects of microorganism decontamination on cotton textile samples with different treatments that can be performed independently or as a part of household washing and drying processes. First, we evaluated the antibacterial efficiency of different drying processes, including classical thermal treatment, a combination of thermal and UV treatments (photothermal treatment), and, finally, microwave technology. The second part of the study was carried out by adding saturated steam to the washing drum.

The reason for the evaluation of the selected procedures for the removal of microorganisms from textiles was the outbreak of COVID-19, when Public Health Institutions, due to the lack of personal protective equipment, proposed that households use non-pharmacological measures to prevent the contamination and spread of SARS-CoV-2. Among other things, these institutions also proposed the use of home-made textile face coverings [45,46]. Unfortunately, the recommendations on the care and disinfection of face coverings were not supported by the results of the studies, and, at the same time, it was almost impossible to implement and evaluate the effects of disinfection in the household environment.

### 2.1. Thermal Treatment

Textile bioindicators, located on a holder in the middle of the drying chamber, were exposed to thermal treatment under different conditions in a laboratory dryer. The samples were dried for a period of between 30 and 90 min at 74 ± 0.71 °C, 82 ± 0.51 °C, and 91 ± 0.23 °C. The degree of *S. aureus* reduction was calculated based on an evaluation of the initial number of colonies and the number of colonies after the heat treatment. The results, in logarithmic step reductions in *Sa*, are shown in Figure 2.

From Figure 2, we can conclude that the degree of *S. aureus* reduction on the textile bioindicators increased with the temperature and heat treatment time, which was expected. The lowest degree of *S. aureus* reduction was achieved at 70 °C and 30 min (RED = 1.432), while 90 min of heat treatment at 90 °C yielded the highest degree of reduction (RED = 3.115). By increasing the time from 30 to 90 min at 70 °C, the reduction factor increased by 0.51, the same as at 80 °C, while at 90 °C, slight increases were observed to a value of 0.71.

We concluded that the low reduction rates were due to the testing conditions of the drying air, which was dormant (natural convection). Air circulation performs a vital transport function during laundry drying, as it supplies the heat and dissipates moisture from the textile material, which occurred to a minimal extent during the thermal treatment of the textile samples.

Based on the results (not shown), we concluded that the fabric temperature was lower than the temperature of the drying air. Most of the heating energy was used to evaporate the surface water and the water between the threads and heat the fabric surface. The consequence of the low air circulation is reflected in the low degree of reduction in *S. aureus* on the textile swatches and in the temperature regulation in the heating chamber (temperature sensors, heaters, and electronic control system).

Therefore, the thermal treatment did not reach at least a 7 log step reduction in *S. aureus* on the cotton swatches. Despite this, we were interested in determining the conditions of drying in the heating chamber with natural air convection under which it would be possible to achieve the desired reduction in microorganisms. For this purpose, we used the GInaFit freeware software tool.

With the help of the GInaFIT V1.7 software, various mathematical models for the prediction of the inactivation of the microbial population were tested, from a simple log-linear model to the Weibull model. The use of the log-linear prediction model, which is a simple first-order model that represents exponential inactivation, gave a coefficient of determination (R^2^) of 0.756. Meanwhile, in the Weibull prediction model, which represents the decline in microbial numbers as a cumulative distribution of heat lethality [47], the R^2^ ranged from 0.9954 to 0.9987 for thermal treatment at 70 to 90 °C. The results are shown in Table 1, while the survival curves of *S. aureus* on textile swatches thermally treated in a heating oven are presented in Figure 3.

As expected, the D_value_ decreased from 33.91 to 21.58 with respect to the temperature increase from 70 °C to 90 °C (Table 1). Based on the Weibull model, a 7 log step reduction would be achieved with heat treatment for 237.37 min at 70 °C, while heat treatment at 90 °C would yield the same reduction after 21.58 min.

Research on the applicability of a prediction model should be expanded to include other temperature-resistant microorganisms typically found on textiles in hospitals and nursing homes, such as temperature-resistant *Enterococcus faecium*. It is also necessary to consider the preparation and treatment conditions (pH, temperature, and moisture content) that could be used to predict the microorganism reduction efficiency of textile hygiene processes more reliably and eliminate long-term testing procedures [48,49,50,51,52].

### 2.2. Photothermal Treatment

A thermogravimetric moisture analyser was used to evaluate the antibacterial efficiency of the combined thermal and UV treatment of cotton swatches contaminated with *S. aureus*. The initial intense emission of UV waves and heat from the halogen lamp above the test samples was followed by the emission of shorter, less-severe waves of light and heat depending on changes in the mass of the dried sample. The treatment process was completed when the difference in mass between successive measurements was less than 0.01%. The drying profiles of the treated samples and the decontamination effects are presented in Table 2 and Figure 4, respectively.

The results (Table 2) show that the reduction in *S. aureus* increased with the temperature of the photothermal treatment; however, the treatment did not inactivate the entire population of *S. aureus* bacteria on the test samples. Thus, the difference between the highest reduction rate and the lowest reduction rate is 2.690 log steps. The highest reduction rate was provided by treatment at 90 °C for 4.008 log steps, while most of the *S. aureus* bacteria survived at 50 °C (6.982 log CFUs/mL). With the increase in the temperature of the photothermal treatment, the treatment time was shortened (Figure 4), and a lower number of surviving *S. aureus* bacteria on test samples was observed.

It is known that the process of drying cotton textiles with hot air has several stages. When heating the fibre by convection, a slight decrease in the moisture content (adhesive water) is characteristic of the initial phase of hot air drying (Figure 4b). This process is followed by the migration of water from inter-fibre spaces to the fibre surface (the hygroscopicity of the fibre surface for capillary water) and a rapid decline in the moisture content (Figure 4a). The formation of a plateau on the moisture content curve indicates the phase when only hygroscopic water remains in the fibre (the equilibrium moisture content, ECM). If hydroscopic water is removed due to an uneven moisture distribution, dehydration occurs, which damages the fibre (a reduction in dimensional stability, stiffness, smoothness, and breaking strength) [53,54,55].

Based on the results, it can be concluded that the microorganism reduction factor lags behind the drying profile in the photothermal treatment. This can be attributed to the phenomenon pointed out by Probstein and Donnarumma that the migration of moisture towards the fibre surface is affected by the surface tension of the water, including the concentration of surface-active agents and the surface potential [56,57]. In our case, however, it should be pointed out that the test samples were impregnated with a suspension of microorganisms before the photothermal treatment, which may have further contributed to the irregular migration of moisture and the inhomogeneous heating of the fibre. This finding is consistent with previous studies [52,58,59].

It is interesting to compare the drying profiles (80 °C) of a cotton sample wetted in water (total water hardness = 2.6 mmol/L) and a cotton sample contaminated with an *S. aureus* suspension (Figure 5). In the initial stages of the photothermal treatment (1–15 s) (Figure 5b), we observed no difference between the drying curves, both samples were heated evenly, and the moisture content did not change significantly. In the second phase of the drying process (Figure 5a), however, differences occurred when the moisture content began to decrease. The moisture content of the sample contaminated with *S. aureus* lagged behind the decrease in the moisture content of the samples soaked with water. This change occurs in the middle of the drying diagram (250 s). The moisture content of the sample contaminated with *S. aureus* decreased faster than that of the sample soaked with water. The decrease in the water content of samples soaked in water was almost linear and ended at 530 s. The reduction in the water content of the sample contaminated with *S. aureus* turned into a concave curve shape and ended at 575 s, which is 45 s later than in the sample wetted only with water. The diagram in Figure 5 shows another difference in the curves, indicating a change in the moisture content depending on the drying time. On the moisture content curve of the test samples soaked with *S. aureus*, several “stepped” sites can be observed. The occurrence of “stepped” parts of the curves was also observed in studies of the sorption properties of cotton samples by tensiometry [60]. These parts were due to the fact that it was difficult for water to migrate into the intermolecular areas of the fibre and inorganic/organic impurities that act as semipermeable membranes and affect the diffusion rate [61]. The observed phenomenon coincides with the findings of Hseih, Takashima, and Sanders, who reported that *Staphylococci spp.* adhered to cotton, PES, and blends of these materials to a greater extent compared with other microorganisms. They also stated that the fabric’s water absorbency and saturation seem to increase the interactions between bacterial cells and the fibre [62,63,64].

### 2.3. Microwave Treatment

Microwave treatments were conducted in two series of tests (samples with a high moisture contest and samples with a low moisture content). In the first series of microwave treatments, test samples contaminated with an *S. aureus* suspension were used, followed by 2 h of drying in a safety chamber at room temperature. The moisture content of the test samples before drying was 85%, while after 2 h of drying in a safety chamber (22 °C, 30% RH) the moisture content was slightly lower (73%).

In the second series of tests, we were interested in how the lower moisture content of the bio contaminated textile samples would affect the antimicrobial efficiency of the microwave treatment, which is a common situation when preparing textile bioindicators for long-term use. Application of the suspension and 2 h of drying of the textile samples in a safety chamber were followed by additional drying in Petri dishes supported in a safety chamber at a temperature of 22 °C and an RH of 30%, followed by 24 h of storage in a refrigerator (8 °C, 40% RH). The average moisture content of the textile samples after 24 h of drying and 24 h of storage was 12%.

The samples from the first and second sets were treated in a microwave oven at 460 W for different treatment times (from 15 to 90 s). Because of the known non-uniformity of the microwave power distribution in the oven cavity (which causes localised hot or cold spots), all contaminated textile samples were cured at the same location in the oven cavity, which was determined as described in [65,66]. The results are shown in Figure 6.

It can be concluded that the population of surviving *S. aureus* bacteria decreased by 1.36 log CFU/mL after 24 h of drying and 24 h of storage (a reduction from 7.497 to 6.134 log CFU/mL). Figure 6 shows that the reduction factor increased with the microwave treatment time, where a complete reduction in *S. aureus* (RED> 7.50 log steps) was achieved after 60 min. In contrast, the microwave treatment time at 460 W of test samples with 12% moisture content did not affect the increase in the RED factor and achieved a maximum reduction at 90 s of 1.33 log steps. The small effect on decontamination can be attributed to the low moisture content level and the small amount of heat generated during the microwave treatment.

The advantage of microwaves, which makes them attractive for use in textile hygiene processes, is their ability, under suitable conditions, to produce rapid and uniform heating throughout the material exposed to them. Microwave heating utilises the effect of a varying and high-frequency voltage on non-conductive (dielectric) material, which creates cyclic strains due to rapid changes in vibrational and rotational movements in the molecules. Under normal conditions, polar molecules of water, the most common polar molecules in textile fabrics, are oriented randomly. When wet textile fabric is exposed to alternating electromagnetic microwave energy, the polar molecules of water change their polarity rapidly and continuously and attempt to align themselves with the changing field (this is also known as the dipole heating mechanism). Friction thus arises from the molecules heating up and finally evaporating. The physics of this phenomenon are such that heating takes place only inside the wet parts of the fabric, and the heat is the most intense where the moisture content is the highest [32].

We also performed microwave tests on contaminated samples in the same way as Chen [34]; i.e., wetting with a suspension of *S. aureus* (130% moisture content), and then, without 2 h of drying, processing the samples with microwaves immediately. We found that microwave treatment of samples at 700 W for 30 s yields a complete reduction in *S. aureus* (RED> 7.50 log steps), which is desired. Less desirable, however, is the finding that there was damage to the cotton specimens (Figure 7). The visual observation of a change in the cotton swatches’ colour from pastel, before the treatment, to yellow-brown, after the microwave treatment, indicates the consequences of thermodegradation of the cotton fibre.

### 2.4. Steam Treatment

The treatment of contaminated textile samples was performed in a drum washing machine to which an external steam generator was connected. The injection of saturated steam (100–103 °C) into the gently agitated washing drum allowed for the homogeneous penetration of steam into all parts of the base load in the washing drum.

First, the temperature profile of the cotton base load was determined depending on the amount of steam added to three loads (30.77% (2.0 kg), 53.85% (3.5 kg), and 100% (6.5 kg)). Figure 8 shows that a 2.0 kg cotton base load reached 60 °C in 20 min by injecting saturated steam and cooled to 55 °C in 7 min. By adding steam at 30-min intervals, it was possible to maintain the temperature of 2.0 kg of ballast between 55 and 65 °C. A total of 3.5 kg and 6.6 kg of cotton ballast heated up significantly less and more slowly. Thus, 400 g of saturated steam was found to heat a 3.5 kg base load in 30 min at 45 °C, while, with 6.5 kg of cotton ballast, the initial temperature increased over the 30-min period by only a few degrees and did not exceed 20 °C. In preliminary research, we found that, for the sustainable and homogeneous heating of ballast, it is necessary to rotate it with a gentle agitation rhythm (3/12/52) and inject saturated steam continuously (400 g/30 min). These findings influenced our decision to carry out the subsequent phases with a 2.0 kg ballast load and processing times longer than 30 min. The results of the influence of the steam treatment in the laboratory washing machine on the reduction in *S. aureus* are shown in Figure 9.

Based on the results (Figure 9), we can conclude that the saturated steam treatment time affects the increase in the S*. aureus* reduction factor. A noticeable decrease in the population of surviving *S. aureus* bacteria on the textile swatches occurred after 30 min of treatment, while a complete reduction in *S. aureus* microorganisms was achieved after 60 min of treatment with saturated steam.

In the steam treatment, the moisture content in the base load increased from an initial 8.5% to 40.81%, 61.46%, and 85.74% after 40, 60, and 90 min, respectively (Figure 10). Saturated steam treatment could function as a stand-alone textile decontamination programme or a pre-programme in a washing programme. In the case of a combination with a washing programme, it would be possible to reduce the laundering machine’s consumption of water significantly, as the base load will be wetted mainly by the pre-steam treatment.

## 3. Materials and Methods

We carried out procedures for the preparation of textile bioindicators, which, in the subsequent step, were treated thermally, photothermally, with microwaves, and with saturated steam for the purpose of evaluating the reduction efficiency of the treatments. A heating and microwave oven, a moisture analyser, a laboratory laundering machine, and a steam generator were used during the research. The bacterial reduction efficiency was measured and evaluated in accordance with ISO, EN, IEC, SIST, or VAH (The Association for Applied Hygiene (D)) requirements, specifications, or guidelines. All experiments were repeated three times.

### 3.1. Materials

Standard cotton fabric (SCF) was used as a microorganism carrier, and accompanying patterns were used to identify cross-contamination during the steaming process. In the research phase, when the effect of steam on decontamination was studied, a base load was used, consisting of IEC T11 cotton sheets, IEC T13 pillow cases, and IEC T12 towels. The characteristics of the SCF and the cotton base load items met the ISO, EN, and DIN standards and are shown in Table 3.

Before inoculation, the SCF was rinsed, dried, cut into swatches 1 m^2^ in size, autoclaved, stored, and protected against recontamination, as is required in DIN ISO 2276:2016, IEC PAS 62,958:2015, and [67]. The ballast load was laundered in the washing machine and then dried in air, as defined in EN 60,456:2016.

The inlet water characteristics met the EN60,456:2016 standard (total water hardness (TWH), 2.5 ± 0.2 mmol/L; pH, 7.5 ± 0.2; T, 15 ± 2 °C).

All used textile materials and the IEC A* detergent were supplied by WFK Testgewebe GmbH (D).

### 3.2. Preparation of Textile Bioindicators

Textile bioindicators are used to determine the microorganism reduction efficiency of different treatments according to SIST EN 16,616:2015, IEC PAS 62,958:2015, and [67]. SCFs were used as carriers for the bacterial cultures of *Staphylococcus aureus DM 799*, which was obtained from the DSMZ German Collection of Microorganisms and Cell Cultures (D). The cotton swatches in the Petri dish were inoculated with 100 µL of artificial sweat that met the ISO 105-E04:2013 standard, as a substrate for simulating human excrement, and left to dry overnight. Sweat was chosen as it was previously found [68,69] that it can act as a substitute for defibrinated sheep blood. In the next step, 100 µL of a bacterial test suspension solution, previously prepared from stock culture as the first and second subcultures, was inoculated onto each cotton carrier. The carriers were then dried at room temperature in open Petri dishes for 2 h in a laminar flow cabinet and then used immediately or stored in the freezer. The initial concentration of bacteria on the cotton swatches was assessed by serial 10-fold dilutions and viable plate counting using Baird-Parker agar and found to lie between 10^7^ and 10^8^ per cotton piece.

### 3.3. Thermal, Photothermal, and Microwave Treatments

#### 3.3.1. Drying and Heating Chamber

Textile bioindicators were treated thermally in an E-28 drying and heating chamber from Binder (D) at different temperatures and for different lengths of time. The main characteristics of the used chamber are a nominal power output of 0.8 kW, an interior chamber volume of 28 L, and dimensions of 400 mm × 280 mm × 250 mm. In one thermal treatment cycle, 5 pieces of the textile bioindicator, lying on a tray in the centre of the chamber, were processed. The air temperature during the thermal treatment was measured by a U12 Hobo single-channel temperature logger (Onset Computer Corp., Bourne, MA, USA) that can record up to 43,000 measurements. After the thermal treatment, the data were transferred to a PC and processed with MS Excel. The thermal treatment of the textile bioindicators was followed by an evaluation of the reduction in the population of the microorganisms in an ordinary way.

#### 3.3.2. Moisture Analyser

A moisture analyser was used to determine the moisture content and photothermal inactivation of microorganisms. The basic principle of the analyser’s operation is the rapid heating of a sample by heaters positioned above it and, consequently, the warming of the sample and the vaporisation of moisture. A precision electronic system determines the sample’s weight continuously during the drying process. The process is completed when the system does not detect a difference in mass between consecutive measurements.

Usually, a moisture analyser device consists of a halogen lamp in which a mercury arc discharge is fired, a reflector system, a cooling system, and an electrical supply. An ordinary halogen lamp is composed of a quartz tube that contains mercury within an inert atmosphere. The lamp’s body is made from quartz, ensuring that the UV energy has maximum transparency. The quartz body can resist an inner temperature of up to 1100 °C. Vacuum-tight tubes with sealed-in electrodes on both ends are used for the discharge of gas. The filling consists of an ignition gas (mostly argon) and liquid mercury spheres. A glow discharge is caused if the voltage above the cold electrodes is sufficient. When the electrodes are heated up, electrons are released from the cathode. This causes rapid growth in the number of electrons via shock ionisation of the filling gas. There is an arc discharge in the noble gas. Electrical energy is transferred to the light arc via the kinetic energy of the electrons. Impulses distribute the energy, the tube is heated up, and the mercury evaporates. The relative spectral density (RSD) of a halogen quartz glass heater is shown in Figure 11.

Just as important as the lamp itself is the system of reflectors that reflect the radiated light sideways or backwards onto the object. Using mirrors, the complete spectrum of short waves, visible light waves, and IR light waves are directed onto the object. The cooling of the halogen lamp with air is the responsibility of the control system (temperature sensors, microelectronic components, and a computer), which is the basis for the accurate control and fine-tuning of drying process conditions.

The KERN DBS 60-3 moisture analyser uses a thermogravimetric method. The sample is weighed before and after heating, and the moisture content in the material is determined by checking the difference. The moisture analyser’s main characteristics are a power output of 400 W, a temperature range from 50 °C to 200 °C in steps of up to 1 °C, a 25 mm distance between the heater and the sample, and a sample plate made of fi90mm aluminium. The moisture content in the textile sample was calculated using Equation (1):MC = (W_n_ − W_0_)/W_0_·100(1)
where MC is the moisture content (%), W_0_ is the initial weight of the sample before the drying step (g), and W_n_ is the weight of the sample (g) after the drying step.

#### 3.3.3. Microwave Oven

Microwave curing was performed with an MO-17 ME microwave oven from Gorenje (SLO), which people use in their daily life to heat or warm food. The samples of textile bioindicators were microwave-cured at 460 W (cavity dimensions, 315 mm × 199 mm × 294 mm; capacity, 17 L) and a frequency of 2450 MHz over different lengths of time.

Details on the fundamentals of microwave heating and the construction and main parts of a household microwave oven can be found in [29].

### 3.4. Steam Treatment

The steam treatment procedures were conducted in a W365H laboratory washing machine from Electrolux (S) with a capacity of 6.5 kg, a drum volume of 65 L, and the possibility of programming the same mechanical action and duration using the programming controller Clarus Control and the PML Laundry Program Manager software, both from Electrolux (S), for customised laundering or drying procedures. The saturated steam needed for the treatment was produced by a 2331 Veit steam generator (D) (power output, 2.2 kW; steam output, 2.8 kg/h; boiler capacity, 5.9 L), which was connected to the washing machine. Figure 12 shows the scheme for the steam treatment system.

The steaming procedure began with loading the washing machine with a base load (2.5 or 3.0 kg) of textile bioindicators and SCF inserted into different cotton laundering bags (5 pcs/steaming cycle), followed by rotation of the washing drum (gentle drum agitation, 3/12/52) and the injection of saturated steam through a nozzle (solenoid valve; T= 103 °C; p= 1.2 bar; spec. enthalpy= 2684 kJ/kg; steam connection= 9.5 mm). Treatments were performed under different conditions, lengths of time, and amounts of injected saturated steam.

Before each test run, the base load was hot-washed in a laundering machine (cotton; regular agitation; MW= 10 min/90 °C; RI1 and 2) with the addition of IEC A* detergent (base; SPT; TAED), dried, and stored prior to further use according to IEC PAS 62,958:2015.

Prior to each test run, the washing machine was conditioned. For that purpose, a self-cleaning programme was used, consisting of a main washing phase (45 min/85 °C) with a higher amount of water and four cold rinsing phases to cool down the laundering machine.

The electricity consumption was measured using the PowerSense Energomonitor (CZ) application. The temperature profile was measured by a U12 Hobo temperature logger placed among the base load items.

### 3.5. Evaluation of the Microbial Reduction

The concentration of microorganisms was evaluated on the cotton swatches during the incubation process and after the decontamination procedures. After each treatment, textile bioindicators were placed into a saline solution to release the cotton swatches. *CFU* was assessed by serial 10-fold dilutions and plating on selected agars. The average *CFU* after the incubation period (48 h/37 °C) was used to determine the reduction efficiency using Equation (2):RED = log (N_0_/N)(2)
where RED is the colony reduction factor efficiency, N_0_ is the average CFU on the cotton swatches before the treatment procedure, and N is the CFU on the cotton swatches after the treatment procedure.

The usual limits for counting bacteria on agar plates are between 15 and 300; however, in SIST EN 16616:2015 and IEC 62958:2015 a deviation of 10% is accepted, so the counting limits were 14 and 330.

According to SIST EN 16616:2015, a chemical–thermal textile disinfection process is considered effective when a reduction rate of greater than 7 log steps is reached.

The microorganism inactivation rate caused by a treatment is expressed as the decimal reduction time D_value_, which is used primarily in the food industry as a safety and quality evaluation parameter for thermal and nonthermal processes. It is known that the decimal reduction time is dependent on the temperature, the type of microorganism, and the composition of the medium containing the microorganisms [70]. The D_value_, the time required to reduce the initial population of microorganisms by 90%, was calculated using Equation (3)
D_value_
*=* t/(log N_0_ − log N_t_)(3)
where N_0_ is the initial population of microorganisms (CFU/mL), N_t_ is the surviving population of microorganisms (CFU/mL), and t is the exposure time (min).

The GInaFiT V 1.7 freeware add-in for the MS Excel fitting tool, developed to test different types of microbial survival models on user-specific experimental data, was used in the research [71]. Due to the preliminary test results, the Weibull model was selected for the accurate prediction of microorganisms’ inactivation.

## 4. Conclusions

The present study focused on less-investigated procedures for removing microbes from household textiles.

Gram-positive *S. aureus* is often selected as a test strain for the evaluation of disinfection properties of laundering procedures, detergents, and disinfectants in accordance with the recommendations of relevant standards. Therefore, it is no surprise that *S. aureus* was used instead of the COVID-19 virus for the development and evaluation of face mask disinfection procedures and agents at the beginning of the COVID-19 pandemic. In later studies, isolates of SARS-CoV-2 were used [17,72,73]. Based on our results, we can conclude that heat treatment in a convection heating chamber with an increased temperature and a longer treatment time affected the increase in the reduction factor but did not ensure a complete reduction in *S. aureus* on cotton swatches. The photothermal treatment of bio contaminated samples yielded higher reduction efficiency, but even this did not ensure the complete inactivation of the bacteria. In contrast, the microwave treatment of bio contaminated textile swatches, containing higher moisture contents, at 460 W for 60 s yielded satisfactory bacterial reduction efficiency. Additionally, the treatment of textiles in a household washing machine and the injection of saturated steam into the washing drum with a mild agitation rhythm yielded at least a 7 log step reduction in *S. aureus*.

When evaluating the effects of textile hygiene treatments, we must be aware that textiles are three-dimensional, flexible, and porous structures with specific sorption, dielectric, and heat-insulating properties. The periodic outbreak of epidemics proves that household textiles, caring for and the hygiene of household textiles, and the hygiene of washing/drying equipment play an essential role in preventing the spread of infections and protecting the health of household members. The change in the health situation must be followed by the development of washing and drying techniques for textile materials, but the principles of sustainable development must be considered too.

## Figures and Tables

**Figure 1 molecules-27-03667-f001:**
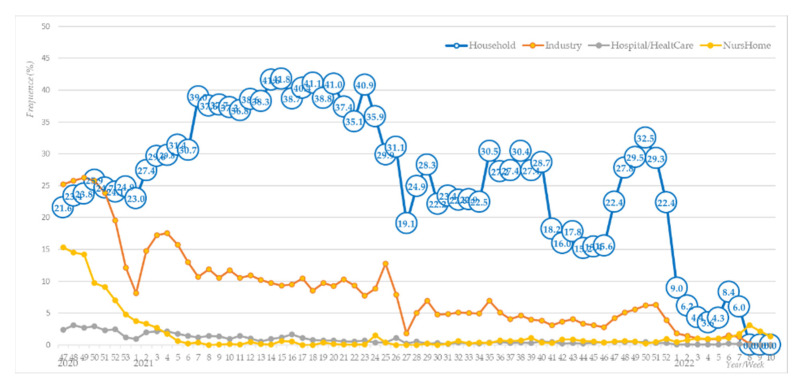
Weekly number of infections depending on the most probable location of virus transmission in the Republic of Slovenia for the period from 1 November 2020 to 19 March 2022 [44].

**Figure 2 molecules-27-03667-f002:**
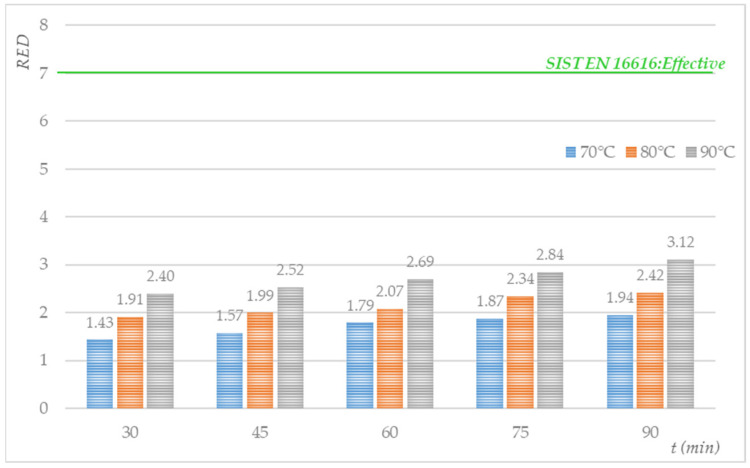
Log step reduction in *S. aureus* on textile bioindicators thermally treated in a heating oven (average N_0_ = 1.47 × 10^7^ CFUs/mL).

**Figure 3 molecules-27-03667-f003:**
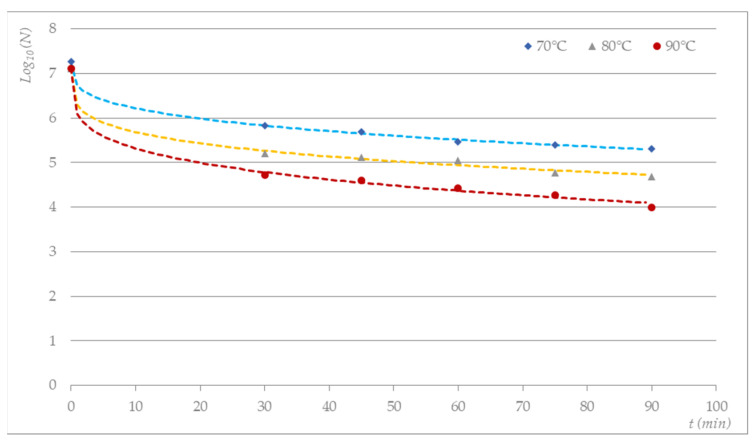
Measured and predicted surviving *S. aureus* microorganisms on textile bioindicators thermally treated in a heating oven based on the Weibull prediction model (GInaFIT v1.7).

**Figure 4 molecules-27-03667-f004:**
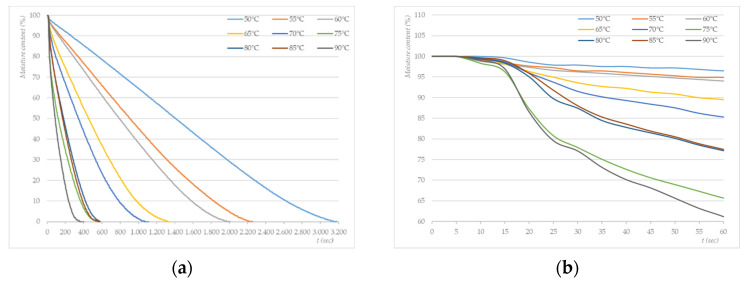
(**a**) Drying profiles of photothermally treated cotton swatches inoculated with *S. aureus* (average N_0_ = 1.18 × 10^8^ CFU/mL); (**b**) magnification of the first 60 s of treatment.

**Figure 5 molecules-27-03667-f005:**
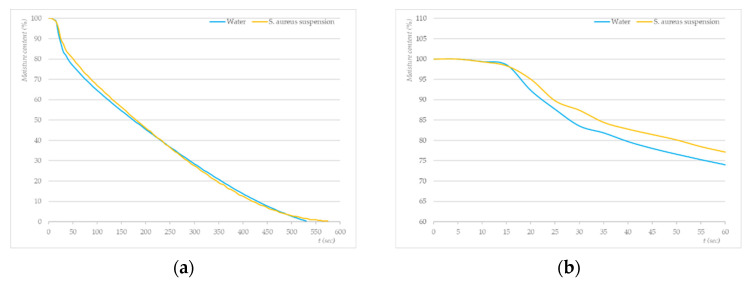
(**a**) Drying profiles of cotton swatches wetted with water (blue line) and swatches inoculated with a suspension of *S. aureus (*N_0_ = 3.67 × 10^7^ CFU/mL, orange line) treated photothermally at 80 °C; (**b**) magnification of the first 60 s of the photothermal treatment.

**Figure 6 molecules-27-03667-f006:**
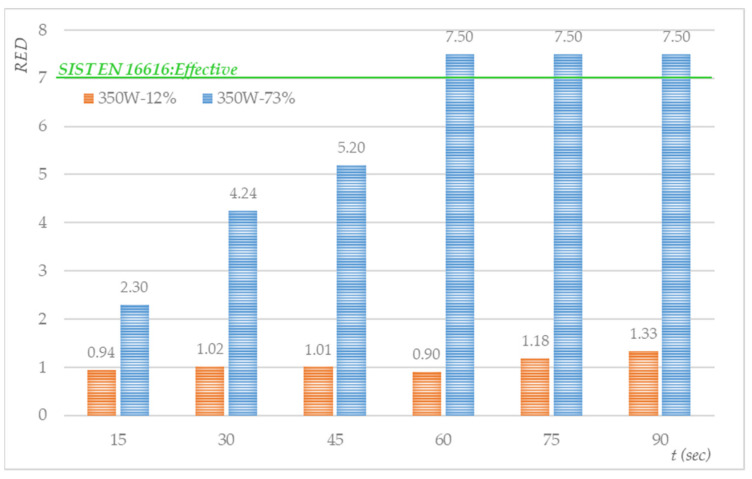
Antimicrobial efficiency of the microwave treatment of cotton swatches inoculated with *S. aureus (*N_0,350W,73%MC_ = 3.14 × 10^7^ CFU/mL, N_0,350W,12%MC_ = 1.36 × 10^7^ CFU/mL).

**Figure 7 molecules-27-03667-f007:**
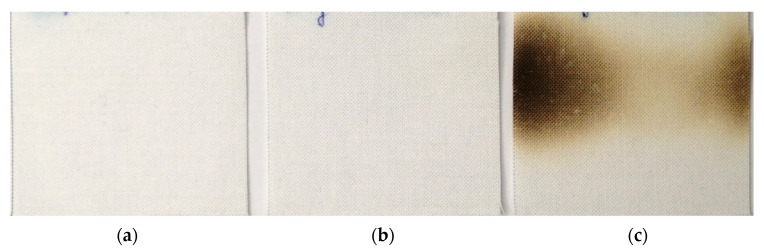
Damage to the cotton swatches after the microwave treatment; (**a**) 700 W, 10 s; (**b**) 700 W, 20 s; (**c**) 700 W, 30 s.

**Figure 8 molecules-27-03667-f008:**
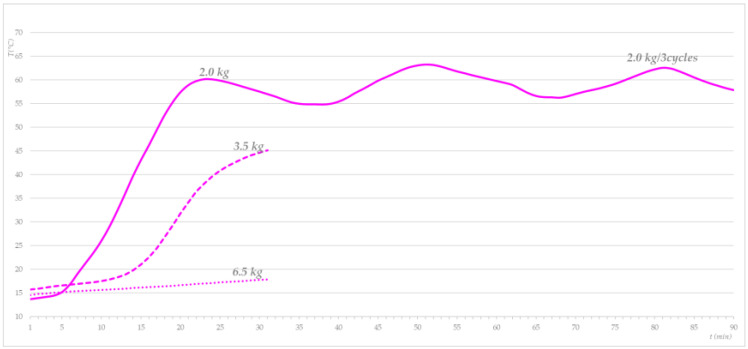
Temperature profile for different cotton base loads treated with saturated steam.

**Figure 9 molecules-27-03667-f009:**
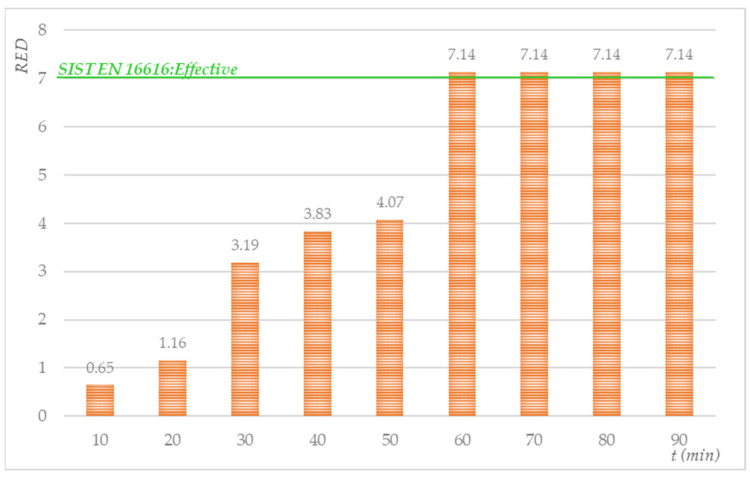
Antimicrobial efficiency of the injection of saturated steam into the drum of a laundering machine containing cotton swatches inoculated with *S. aureus (*average N_0_ = 2.56 × 10^7^ CFU/mL).

**Figure 10 molecules-27-03667-f010:**
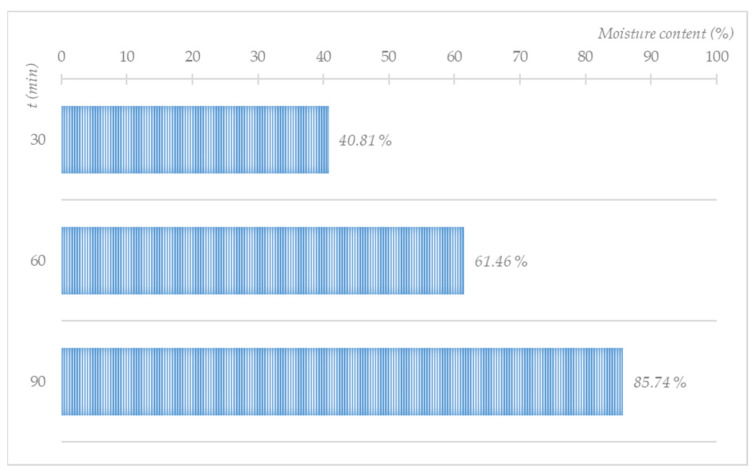
Influence of the treatment with saturated steam (400 g/30 min) on the wetting of a cotton base load (2.0 kg).

**Figure 11 molecules-27-03667-f011:**
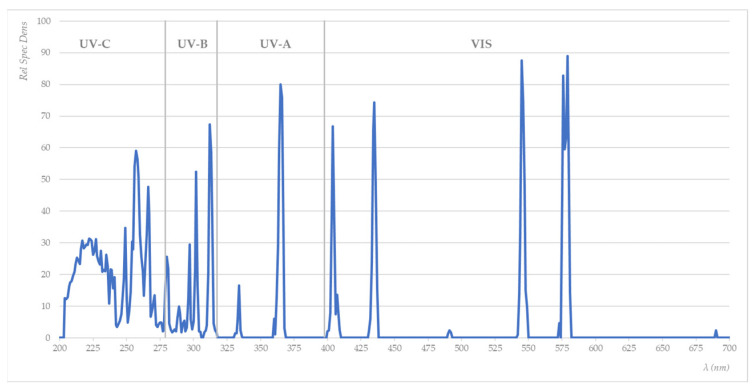
Relative spectral density (RSD) of a halogen quartz glass heater.

**Figure 12 molecules-27-03667-f012:**
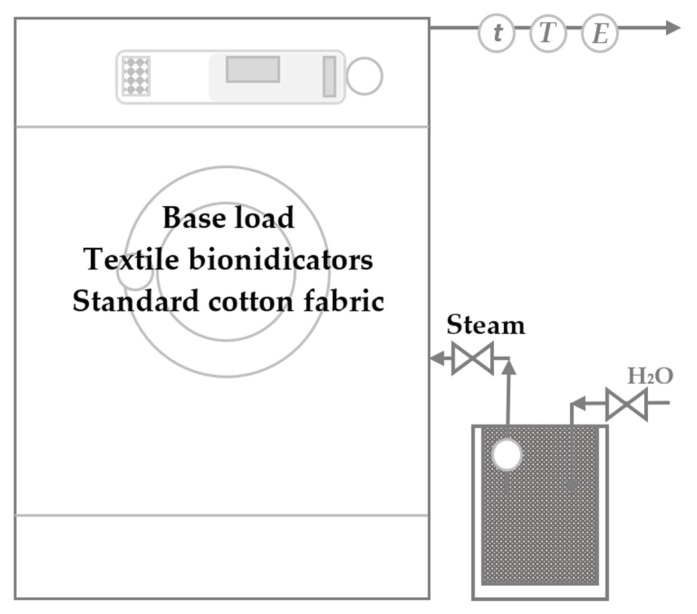
Scheme of the system for the saturated steam treatment.

**Table 1 molecules-27-03667-t001:** The Weibull prediction model parameters for the heat resistance of *S. aureus* on textile swatches thermally treated in a heating oven.

T (°C)	N_0_ (log CFUs/mL)	D_value_ (min)	t_7D_ (min)	RMSSE	R^2^
70	7.25	33.91	237.37	0.034	0.9987
80	7.11	27.28	190.27	0.078	0.9954
90	7.11	21.58	151.06	0.083	0.996

N_0_, initial value of *S. aureus* before thermal treatment; D_value_, decimal reduction time; t_7D_, time needed to reach a 7 log reduction in *S. aureus* at temperature T; RMSSE, Root Mean Sum of Square Errors; R^2^, coefficient of determination.

**Table 2 molecules-27-03667-t002:** Influence of photothermal treatment on the survival of *S. aureus* on textile swatches.

T (°C)	t (min)	N (log CFUs/mL)	RED
-	0	8.301	-
50	53	6.982	1.319
55	37	6.663	1.638
60	33	6.465	1.836
65	26	6.029	2.272
70	20	5.664	2.637
75	14	4.655	2.910
80	10	4.193	3.372
85	7	3.756	3.809
90	4	3.556	4.008

T, treatment temperature; t, time of treatment set by the moisture analyser (diff <0.01% between two continuous measurements); N, average value of the surviving *S. aureus* after exposure to the treatment; RED, reduction factor.

**Table 3 molecules-27-03667-t003:** Characteristics of the used fabric and base load items.

Parameter	Standard Cotton Fabric	Base Load		
Standard	DIN ISO 2267:2016	SIST EN 60456:2016, IEC PAS 62958:2015		
Form	Fabric	Bed sheet	Pillowcase	Towel
Fibre composition	100% cotton	100% cotton		
Density/Warp	27 threads/cm, 295 dtex	24 threads/cm, 330 dtex		24 threads/cm, 330 dtex
Density/Weft	27 threads/cm, 295 dtex	24 threads/cm, 330 dtex		12 threads/cm, 970 dtex
Mass	170 g/m^2^	185 g/m^2^		220 g/m^2^
Weave	Plain	Plain		Huckaback
Finish	Desizing, boiling, singeing, bleaching			
Colour	C*_D65/10_ = 0.65, h_D65/10_ = 98.62, WI_CIE_ = 71.69	Not defined		

## Data Availability

Not applicable.

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
