# Peer review of "Decontamination Efficiency of Thermal, Photothermal, Microwave, and Steam Treatments for Biocontaminated Household Textiles"

_molecules, 2022, doi:10.3390/molecules27123667_

Round 1

Reviewer 1 Report

Dear Authors,

Overall, I like the paper. I have read the manuscript and offered some comments. At first instance, the manuscript seems interesting, but it lacks coherence after reading the complete manuscript. The following recommendations can be implemented:

  • Please keep your sentences smooth and easy to read. There are minor grammatical errors and some inconsistent spacing. Please revise.
  • Figures do not have consistency in style. Figures’ legends and titles are not readable (e.g., Figs 5 (a) and (b), 8 and 11) 
  • The introduction section may be condensed by approx. 30%.
  • Line 423, plain weave?
  • I would use a Table for section 3.1 to show material features (structure and properties)

The authors used three different base loads: sheet IEC T11, pillow cases IEC T13, and towels IEC T12. They have the same fiber content. Do they have the same structure? If not, the effect of the fabric structure is not discussed. 

Is there any logic why only 100% cotton has been used as it is customary to blend synthetic fibers for bed sheets and pillow covers?

Section 3.4 is not clear; steam characteristic? Nozzle? Steam pressure? Steam temperature? How was the steam injected? How much energy is delivered by the steam?

Kind regards, 

Reviewer 2 Report

Given the situation related to today's problems caused by the action of microorganisms on human health, the research within the work is well-directed but requires the necessary supplementation.

The authors mention the reason for the evaluation of the selected microorganism removal procedures from textiles was also, partly, in the outbreak of the Covid-19 problem related to SarsCov-2 virus and the paper does not clearly show the impact of treatment on hygienic properties of textiles on viruses as well as on gram-negative bacteria and fungi. Please expand on the results of the impact of textile care on the presence of viruses or theoretically clarify the correlation between the results shown for gram-positive bacteria (S. aureus) in relation to viruses of course if possible.

line 28 - lowercase letter – laundering

line 48 - put 2 in the index - so CO2

line 149 - unclear Fig. X ??

line 199 - Sa. throughout the text put in italics

line 238 - unify the use of the abbreviation for the decimal reduction time throughout the text

line 282 - decrease breaking strength without and

line 435 - list the names of detergents used in the research

from line 505 to line 519 - it is not necessary to state the mode of operation of the microwave oven, but cite the literature that describes it

line 564 - use full name Decimal reduction time

Figures 1, 4, 5 and 8 are very poor resolutions

Check and even out fonts throughout the paper.
